# Effects of Whey Peptide Supplementation on Sarcopenic Obesity in High-Fat Diet-Fed Mice

**DOI:** 10.3390/nu14204402

**Published:** 2022-10-20

**Authors:** Gahyun Lim, Yunsook Lim

**Affiliations:** Department of Food and Nutrition, Kyung Hee University, Seoul 02447, Korea

**Keywords:** sarcopenic obesity, resistant exercise, whey peptide, protein degradation, adipogenesis, adipocyte differentiation

## Abstract

The incidence of sarcopenic obesity gradually increased in parallel with the aged population. This research examined the effects of whey peptide (WP) supplementation with/without resistant exercise (RE) on sarcopenic obesity. Male 8-month-old C57BL/6J mice were fed a control diet (10 kcal% fat) or a high-fat diet (60 kcal% fat) for 8 weeks. High-fat diet-fed mice were randomly divided into four groups: obesity control group (OB), RE (RE only), WP (WP only), and WPE (RE and WP). WP supplementation (1500 mg/day/kg B.W.) gavage and RE (ladder climbing, five times weekly, 8–10 repetitions, 10–20% B.W. load) were conducted for an additional 8 weeks. Protein and mRNA levels of markers related to energy, protein, and lipid metabolism were analyzed in skeletal muscle and adipose tissue by one-way analysis of variance (ANOVA). WP supplementation regardless of RE significantly suppressed the increasing fat mass (*p* = 0.016) and decreasing lean mass (*p* = 0.014) and alleviated abnormal morphological changes in skeletal muscle and adipose tissue (*p* < 0.001). In adipose tissue, WP supplementation regardless of RE ameliorated dysregulated energy metabolism and contributed to the reduction in adipocyte differentiation (PPAR-γ (*p* = 0.017), C/EBPα (*p* = 0.034)). In skeletal muscle, WP supplementation regardless of RE alleviated energy metabolism dysregulation and resulted in down-regulated protein degradation (Atrogin-1 (*p* = 0.003), MuRF1 (*p* = 0.006)) and apoptosis (Bax) (*p* = 0.004). Taken together, the current study elucidated that WP supplementation regardless of RE has potential anti-obesity and anti-sarcopenic effects in sarcopenic obesity.

## 1. Introduction

Sarcopenic obesity is a coexistence of sarcopenia and obesity, which is characterized by obesity with low muscle mass and muscle strength [1]. It is common in the aged population and seriously deteriorates the quality of life due to falls and fractures [2]. Metabolic dysregulation, oxidative stress, inflammation, and insulin resistance caused by both aging and obesity result in body compositional changes, eventually leading to sarcopenia-like characteristics [3,4]. According to the Centers for Disease Control and Prevention (CDC) 2020, it has shown that aging and obesity are concurrent phenomena, as indicated by the highest obesity prevalence in middle-aged (age: 35–64 years) adults [5].

In sarcopenic obesity, high fat mass coupled with low lean mass hinders the regulation of energy metabolism in various organs [6]. Dysregulated energy metabolism stimulates adipogenesis in adipose tissue, induces protein degradation and inhibits protein synthesis in skeletal muscle [7]. Although a lot of animal studies regarding sarcopenic obesity have been conducted, there are no clear criteria for the appropriate age of the sarcopenic obese animal model. According to previous studies, sarcopenic obesity was induced by a high-fat diet in different aged mice (the young [8], the middle-aged [9], and the old [10]). As the aged population has a higher chance of sarcopenic obesity that is easily initiated in middle age, the middle-aged animal model was used in this study.

Diverse physical activity and dietary interventions have been implemented to treat sarcopenic obesity. Recent randomized controlled trials (RCT) have demonstrated that physical activity interventions possess an ameliorative effect on sarcopenic obesity by improving body composition and muscle strength [11,12,13]. Several meta-analyses have shown that resistant exercise (RE) is the most effective type of exercise in sarcopenic obesity [14,15,16]. Among dietary interventions, high-protein intake has been suggested as an optimal option by several reviews [17,18]. Adequate protein intake is significant because an imbalance between protein requirement and supply causes a loss of skeletal muscle mass due to the disruption in muscle protein synthesis and degradation dynamics [19]. Moreover, a recent RCT study demonstrated that adequate protein intake reduced muscle loss in aged women with sarcopenic obesity [20].

Whey protein has shown numerous health benefits, including body composition improvement, immunity enhancement, and inflammation and oxidative stress reduction in human studies [21,22,23]. Whey peptide (WP), a hydrolyzed form of whey protein, has a higher absorption rate than whey protein [24] and a variety of biological benefits including anti-obesity [25] and antioxidant [26] effects in vivo and fast recovery from exercise-induced muscle damage in a human study [27]. Although whey protein and WP have similar biological benefits, a recent study demonstrated that WP more efficiently reduced aging-related oxidative stress compared to whey protein [26]. It might be because bioactive peptides have stronger cellular affinity and specificity with a higher absorption rate than proteins [28], which is expected to be more effective for metabolic disorders such as obesity [29].

Many studies have demonstrated the potential of the combination of protein supplementation with exercise on sarcopenic obesity. Several RCTs have demonstrated the possibility of the combination; dietary protein supplementation is a prerequisite for muscle mass increase during RE in frail aged people [30], the combination of heavy RE and protein supplementation significantly improved muscle mass and strength compared to protein supplementation only in healthy aged people, and whey protein supplementation with pre-resistant exercise increased skeletal muscle mass, muscle strength, and physical functional capacity in aged women [31]. However, the combined effect of dietary WP supplementation and exercise intervention on sarcopenic obesity is still uncertain [32]. This study aimed to investigate the molecular mechanism in which WP supplementation and RE ameliorate sarcopenic obesity in high-fat diet-induced obese mice.

## 2. Materials and Methods

### 2.1. Preparation of Whey Peptide

Whey peptide, manufactured by Tatua Co-operative Dairy Company Limited (Waikato, New Zealand), provided by Megmilk Snow Brand Co., Ltd. (Tokyo, Japan), and supplied by Ju Yeong NS Co., Ltd. (Seoul, Korea) is used. Whey peptide was carefully suspended in distilled water and stored at 4 °C.

### 2.2. Animals and Sarcopenic Obesity Induction

Eight-month-old C57BL/6J male mice (*n* = 60) were obtained from Janvier Labs (Rte du Genest, France) and acclimated for one week under standard conditions (temperature: 22 ± 1 °C, humidity: 50 ± 5%, 12 h light–dark cycle). After a week of acclimation, all mice were randomly divided into two groups for sarcopenic obesity induction: a normal diet group (ND; *n* = 12) and a high-fat diet group (OB; *n* = 48). The high-fat diet group was fed a 60% kcal high-fat diet (D12492; Research Diets, New Brunswick, NJ, USA), while the normal diet group was fed a 10% kcal fat control diet (D12450J; matching sucrose to D12492, Research Diets, New Brunswick, NJ, USA) for 8 weeks, which was provided ad libitum. All of the experimental protocols using animals were approved by the Institutional Animal Care and Use Committee of Kyung Hee University [KHSASP-21-247].

### 2.3. Experimental Design

After sarcopenic obesity induction, mice were randomly divided into five groups (*n* = 10): (1) normal control group (CON)—fed with a 10% kcal fat diet and supplemented with distilled water; (2) obesity control group (OB)—fed with a 60% kcal fat diet and supplemented with distilled water; (3) resistant exercise control group (RE)—fed with a 60% kcal fat diet and supplemented with distilled water; (4) whey peptide supplementation treated group (WP)—fed with a 60% kcal fat diet and supplemented with 1500 mg/kg BW of whey peptide; (5) whey peptide supplementation and resistant exercise-treated group (WPE)—fed with a 60% kcal fat diet and supplemented with 1500 mg/kg BW of whey peptide. The experimental design is shown in Figure 1. Whey peptide was orally gavaged every day for eight weeks, specifically 30 min after resistant exercise for exercising groups. Modified resistant training (ladder climbing) tested on rats by Kim and Song [33] was implemented considering the age of the mice. It was performed with a home-made 1 m ladder (1.5 cm grid steps, inclined 85–90°) five times weekly [33] and gradually increased the load up to 20% of B.W. with a maximum of 12 repetitions, as shown in Figure 2. The progressive increase in exercise intensity was executed by adding weight to the animal’s tail and increasing repetitions [34]. Exercising mice were gently motivated to climb the ladder by a soft pat on their body [35]. BW and food intake were measured once a week.

### 2.4. Body Composition Analysis

Body composition was analyzed by dual-energy X-ray absorptiometry (DXA; InAlyzer, Seongnam, Gyeonggi-do, Korea) before and after 8 weeks of interventions. After anesthetized with ketamine and xylazine, each mouse was placed on the scanner bed with the tail and limbs stretched away from the body.

### 2.5. Muscle Strength Test

The grip strength test was performed using a grip strength meter (Grip test package GS3 (25 N); Harvard Apparatus, Holliston, MA, USA). The mice were directed to grasp the sensor bar, and we gently pulled the tail of the animal horizontally with the body at a fixed speed until their forelimbs were released. The highest level of forelimb grip force was recorded, and the average value of six repetitions was used.

### 2.6. Histological Assays

The epididymis fat and gastrocnemius muscle were fixed with 10% formalin solution and were embedded in paraffin. The embedded tissues were cut into 4 µm slides and then stained with hematoxylin and eosin. The stained sections of epididymis fat and gastrocnemius muscle were observed using an optical microscope (Nikon ECLIPSE Ci, Konan, Tokyo, Japan). The mean percentage of myofiber area was analyzed using Image J software (NIH, Bethesda, MD, USA).

### 2.7. Protein Extraction and Western Blot Analysis

The protein extraction of skeletal muscle (gastrocnemius) adipose tissue (epididymis fat) was performed according to a method previously described [36,37]. Protein was quantified using a BCA Protein Assay Kit (Abcam, Cambridge, MA, USA). The following primary antibodies are used: PGC-1α, pAMPK, AMPK, FoxO3a, Atrogin-1, MuRF1, Bax, Bcl-xL, Akt, pAkt (Santa Cruz Biotechnology, CA, USA, 1:200); S6K1 (Cell Signaling Technology, MA, USA, 1:1000); α-tubulin (Sigma-Aldrich, St. Louis, MO, USA, 1:4000); IGF-1 (Abcam, Cambridge, MA, USA, 1:1000); β-Actin (Bethyl Laboratories, Montgomery, TX, USA, 1:500); PCNA (Enzolife science, Farmingdale, NY, USA, 1:1000).

### 2.8. RNA Extraction and Real-Time PCR Analysis

Total RNA was extracted in adipose tissue (epididymis fat) using TRIzol (Takara, Otsu, Japan), and the RNA concentration was assessed by a NanoDrop™ One/OneC Microvolume UV-Vis Spectrophotometer (Thermo Fisher Scientific, Waltham, MA, USA). The cDNA was synthesized from the extracted RNA using a Prime Script RT reagent kit (Takara, Otsu, Japan), and synthesis was used for the real-time RT-PCR analysis with a SYBR Premix Ex Taq II (Takara, Otsu, Japan) using StepOnePlus (Applied Biosystems, Foster City, CA, USA). The utilized primer sequences are displayed in Table 1. GAPDH was used as a reference gene, and relative mRNA expression compared to the control group was normalized.

### 2.9. Statistical Analysis

All the numerical data were expressed as means ± standard error of the mean (SEM). The significance of differences among the groups was assessed by one-way analysis of variance (ANOVA) according to Duncan’s multiple range test using SPSS software (SPSS Inc., Chicago, IL, USA). A significant level of *p* < 0.05 was implemented.

## 3. Results

### 3.1. Effects of WP Supplementation and RE on Diet Intake, Body, Fat, and Muscle Weights, Body Composition, and Muscle Strength

Compared to the CON group, the high-fat diet-fed groups had significantly higher BW and fat weights with lower muscle weights. WP supplementation with RE significantly lowered BW compared to the OB group (*p* < 0.001). Regarding the BW difference, WP supplementation regardless of RE significantly inhibited increasing BW in sarcopenic obesity (*p* < 0.001). In addition, WP supplementation regardless of RE significantly reduced total fat weight (*p* < 0.001) with the most drastic change in epididymis fat (*p* < 0.001). Only WP supplementation with RE significantly increased muscle (gastrocnemius) weight compared to the OB group (*p* < 0.001). After 8 weeks of RE and WP supplementation, the OB group had significantly higher fat mass and lower lean mass compared to those of the CON group (*p* < 0.001). Regarding the differences in body composition, WP supplementation regardless of RE significantly restrained increasing fat mass (*p* = 0.016) accompanied by decreasing lean mass (*p* = 0.014) compared to the OB group. The CON group had significantly higher muscle strength compared to the OB group. Among the high-fat diet-fed groups, only the exercise groups showed stronger muscle strength (*p* < 0.001). The amount of diet consumption was significantly larger in the CON group compared to the OB group and smaller in the WPE group compared to the OB group (*p* < 0.001) (Table 2).

### 3.2. Effects of WP Supplementation and RE on Morphological Changes in Adipose Tissue and Skeletal Muscle

The myofiber area of the CON group was significantly larger than that of the OB group. The WP supplementation regardless of RE significantly increased the mean myofiber area compared to the OB group (*p* < 0.001) (Figure 3A,B). The OB and RE groups had significantly larger mean adipocyte sizes than the CON group. The WP supplementation regardless of RE significantly reduced the size of adipocytes compared to the OB and RE groups (*p* < 0.001) (Figure 3C,D).

### 3.3. Effects of WP Supplementation and RE on Adipokines and Lipid and Energy Metabolism in Adipose Tissue

The mRNA expression level of leptin was significantly different between the CON and OB groups (*p* = 0.008). WP supplementation regardless of RE significantly reduced the mRNA expression level of leptin compared to the OB group. Adiponectin mRNA expression did not differ among the groups (*p* = 0.454) (Figure 4A).

There was no significant difference between the CON and the OB groups in the mRNA expression levels of peroxisome proliferator-activated receptors γ (PPAR-γ) and CCAAT-enhancer-bind proteins α (C/EBPα). WP supplementation regardless of RE significantly down-regulated the mRNA expression level of PPAR-γ compared to the OB group (*p* = 0.017). The WP supplementation itself significantly reduced the mRNA level of C/EBPα compared to the OB group (*p* = 0.034) (Figure 4B).

There was no significant difference in the protein levels of 5’ AMP-activated protein kinase (AMPK) (*p* = 0.059)and pAMPK (*p* = 0.058) among the groups. WP supplementation with RE significantly up-regulated the ratio of pAMPK/AMPK (*p* = 0.007). There was no significant difference in the protein levels of peroxisome proliferator-activated receptor-gamma coactivator 1-alpha (PGC-1α) between the CON and OB groups. WP supplementation itself significantly increased the protein level of PGC-1α compared to the OB group (*p* = 0.010) (Figure 4C).

### 3.4. Effects of WP Supplementation and RE on Protein and Energy Metabolism and Apoptosis in Skeletal Muscle

There was no significant difference in the protein levels of forkhead box O3 (FoxO3a) (*p* = 0.070), ribosomal protein S6 kinase beta-1 (S6K1) (*p* = 0.202), and insulin-like growth factor I (IGF-1) (*p* = 0.711) among groups. There was no significant difference in the protein levels of muscle atrophy F-box gene (Atrogin-1) and E3 ubiquitin–protein ligase (MuRF1) between the CON and OB groups. Only WP supplementation itself significantly down-regulated the protein level of Atrogin-1 compared to the OB group (*p* = 0.003). WP supplementation regardless of RE significantly lowered the protein level of Murf1 compared to the OB group (*p* = 0.006) (Figure 5A).

There was no significant difference in the protein levels of B-cell lymphoma-extra large (Bcl-xL) (*p* = 0.491) and the Bax/Bcl-xL ratio (*p* = 0.751) among groups. The protein level of bcl-2-like protein 4 (Bax) was significantly larger in the OB group compared to the CON group. WP supple mentation regardless of RE significantly down-regulated the protein level of Bax compared to the OB group (*p* = 0.004) (Figure 5A).

There was no significant difference in the protein levels of protein kinase B (Akt), pAkt, and pAkt/Akt between the CON and OB groups. The WP supplementation regardless of RE significantly up-regulated compared to the OB and RE groups. WP supplementation regardless of RE significantly increased the protein level of pAkt (*p* = 0.023) and pAkt/Akt ratio (*p* = 0.002). The protein level of Akt was not significantly different among the groups (*p* = 0.176) shown in (Figure 5B).

## 4. Discussion

Sarcopenic obesity is a multi-organ dysfunctional condition caused by several biological changes including a reduction in lean mass and accumulation in fat mass. The incidence of sarcopenic obesity gradually increased in parallel with the aged population due to declined metabolic rates and physical activity. The current research demonstrated that WP supplementation regardless of RE has the potential to alleviate sarcopenic obesity. As the decline of lean mass and accumulation of fat mass begins in middle age, middle-aged mice were used for better human application [38,39].

WP supplementation regardless of RE significantly inhibited an increase in fat mass and promoted an increase in lean mass, indicating simultaneous anti-sarcopenic and anti-obesity effects in sarcopenic obesity. A recent meta-analysis showed that whey protein intake significantly increased lean body mass [40]. Moreover, WP supplementation regardless of RE significantly inhibited increasing BW and reduced fat weight in sarcopenic obesity. However, only WP supplementation with RE significantly decreased BW and increased muscle (gastrocnemius) weight. The ineffectiveness of WP supplementation itself on muscle weight implies that physical exercise is a prerequisite for increasing muscle weight in sarcopenic obesity. In addition, the ineffectiveness of RE itself on muscle weight is expected to be because of the inadequate intensity of RE. It is expected that the combination of WP supplementation and RE stimulated an increase in muscle weight and assisted in weight loss. Both WP supplementation and RE did not change diet intake. The differences in protein and nitrogen intake among groups were not corresponding with the lean mass changes, considering additional whey peptide supplementation and diet intake. It could be interpreted that body composition may be influenced by not only diet but other factors such as gender, aging, behavior, physical activity, and hormones [41]. Altogether, WP supplementation can possibly be an effective modulator of body composition in sarcopenic obesity, but additional exercise is recommended.

In sarcopenia, loss of muscle function, morphological changes, and conversion of muscle fibers and shrinkage are closely associated with the loss of muscle mass that accelerates obesity [42]. Adipocyte hypertrophy induces the dysfunction of adipose tissue, which contributes to metabolic deterioration in obesity [43]. WP supplementation regardless of RE ameliorated abnormal morphological changes in both adipose tissue and skeletal muscle caused by alteration of body composition under sarcopenic obese condition. However, muscle strength was only enhanced in exercise groups, implying that RE promoted an increase in muscle quality. This result corresponds with a recent RCT [44] which suggested that the effects of whey protein or peptide supplementation on muscle strength need further investigation. Thus, the importance of adequate protein intake in reducing abnormal morphological changes and metabolic dysfunction in sarcopenic obesity can be inferred.

Sarcopenic obesity causes energy metabolism dysregulation in both adipose tissue and skeletal muscle [7]. Biological changes in adipose tissue and skeletal muscle directly affect each other, since both are interconnected endocrine organs, forming bidirectional inter-organ crosstalk [45]. Phosphorylated AMP-activated protein kinase alpha (pAMPKα), which is a crucial factor for energy homeostasis in adipose tissue [46] inhibits adipocyte differentiation [47] and adipogenesis [48,49]. Moreover, pAMPKα interacts with PPAR-γ coactivator 1 alpha (PGC-1α), which suppresses adipogenesis via morphological changes in white adipose tissue [50]. In the present study, both WP supplementation itself demonstrated by PGC-1α and WP supplementation with RE represented by pAMPK/AMPK improved energy metabolism homeostasis in sarcopenic obesity [51]. It is expected that WP supplementation regardless of RE significantly reduced fat mass and fat (epididymal) weight by inhibiting adipocyte differentiation demonstrated by PPAR-γ and C/EBPα under the activation of PGC-1α and AMPK in sarcopenic obese condition [50]. Adipose tissue maintains energy homeostasis itself and that of other organs by secretion and communication [52].

In adipose tissue, fat mass accumulation and energy metabolism dysregulation induce aberrant adipokines production, such as hyperleptinemia [53]. Adipokines such as leptin and adiponectin perform various functions in other organs, including the metabolism of skeletal muscle [54,55,56]. In addition, a study indicated that leptin is a linkage between visceral obesity and sarcopenia [57]. WP supplementation regardless of RE significantly alleviated hyperleptinemia by lowering fat mass as indicated by the normalized mRNA expression level of leptin in sarcopenic obesity. It is expected that alleviated energy metabolism dysregulation and hyperleptinemia by WP supplementation regardless of RE contributed to both anti-obesity and anti-sarcopenic alterations under a sarcopenic obese state [58]. Thus, it can be inferred that WP supplementation regardless of RE encouraged a homeostatic control of energy balance, which ultimately reduces fat mass in sarcopenic obesity.

As energy metabolism is closely related to protein metabolism in the skeletal muscle [59], reduced muscle weight or lean mass contributes to energy metabolism dysregulation in skeletal muscle [60]. In skeletal muscle, phosphatidylinositol-3-kinase (PI3K)/protein kinase B (Akt) is a muscle metabolism regulator that induces muscle protein synthesis through the phosphorylation of protein S6 kinase 1 (S6K1) and eukaryotic translation initiation factor 4E (elF4E)-binding protein 1 (4E-BP1) [61]. On the other hand, phosphorylated Akt inhibits protein degradation through inhibition of Muscle RING finger 1 (MuRF1) and muscle atrophy F-box (MAFbx/Atrogin-1), which are muscle-specific E3 ubiquitin ligases mediated by forkhead box O3a (FoxO3a) [62]. In this line, it can be suggested that WP supplementation regardless of RE preserved lean mass by suppressing adverse protein metabolism represented by Atrogin-1, MuRF1, and Bax via the normalization of energy metabolism shown by pAkt/Akt in sarcopenic obesity. Thus, WP supplementation regardless of RE effectively protected lean mass and suppressed fat mass accumulation via improving energy metabolism homeostasis in sarcopenic obesity.

A previous study demonstrated that the consumption of silk peptide is effective in anti-obesity, protein synthesis, and inhibition of muscle degradation in obese mice [47,63]. The WP supplementation used in this study contains about 20% of branched-chain amino acids, which regulate protein turnover [64]. However, we found that WP supplementation regardless of RE did not induce protein synthesis in the skeletal muscle, which is in accordance with unchanged muscle weight. It may be explained by differences in amino acid composition, as silk peptide contains a high percentage of glycine (33.1%) and alanine (28.1%), both of which have shown muscle-protective effects [65,66]. Therefore, we concluded that WP supplementation is mainly activated on the inhibition of protein degradation in the skeletal muscle rather than the promotion of protein synthesis in sarcopenic obesity. Moreover, RE also did not show an increase in protein synthesis-related maker. It is expected that the differing effects of exercise on muscle metabolism according to its type, duration, and intensity [67] and the production of ROS during resistant exercise [68] suppressed protein synthesis in skeletal muscle. In addition, a study demonstrated that progressive lean mass loss blunted the anabolic metabolism (protein synthesis, S6K1) after exercise in human skeletal muscle [69]. Thus, it is expected that protein synthesis in skeletal muscle was not induced by RE due to insufficient protein synthesis in skeletal muscle.

Further studies focusing on the effects of diverse protein supplementation and exercise on various organs will help understand sarcopenic obesity. Particularly, the effects of dietary and exercise interventions in bidirectional interorgan crosstalk between adipose tissue and skeletal muscle need to be fully investigated for the prevention or treatment of sarcopenic obesity [70]. There is a limitation in this study, which is the absence of a whey protein-supplemented control group. Therefore, further research comparing whey protein and peptide supplementation in sarcopenic obesity is needed to clarify the observed effects, clinical application, and deeper interpretation of the mechanism. Considering the dose conversion ratio [71], dietary reference intake, and RCTs related to the effects of protein supplementation in sarcopenic obese patients [20], an additional 10–20 g of WP supplementation is recommended for both sarcopenic obese and normal humans.

## 5. Conclusions

To summarize, WP supplementation regardless of RE ameliorated sarcopenic obesity by improving body composition and normalizing pathophysiological changes that occurred in both adipose tissue and skeletal muscle. Relatively reduced fat mass and increased lean mass normalized energy metabolism dysregulation in adipose tissue and skeletal muscle. Moreover, WP supplementation regardless of RE effectively inhibited adipocyte differentiation in adipose tissue and protein degradation in skeletal muscle. Taken together, WP supplementation regardless of RE could be helpful in a therapeutic approach to sarcopenic obese animals and humans.

## Figures and Tables

**Figure 1 nutrients-14-04402-f001:**
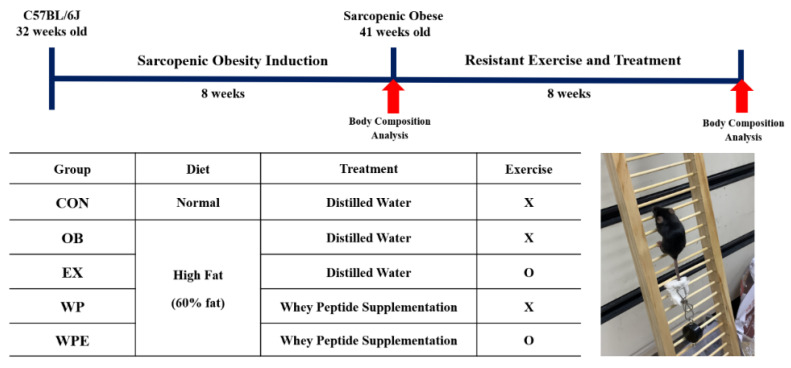
Experimental design. X: non-exercise; O: exercise.

**Figure 2 nutrients-14-04402-f002:**
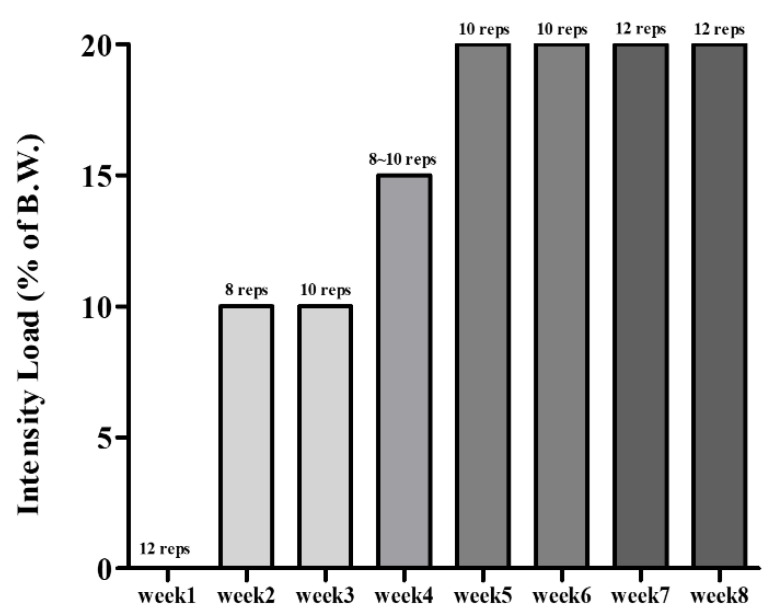
The incremental intensity load applied to the resistant exercise.

**Figure 3 nutrients-14-04402-f003:**
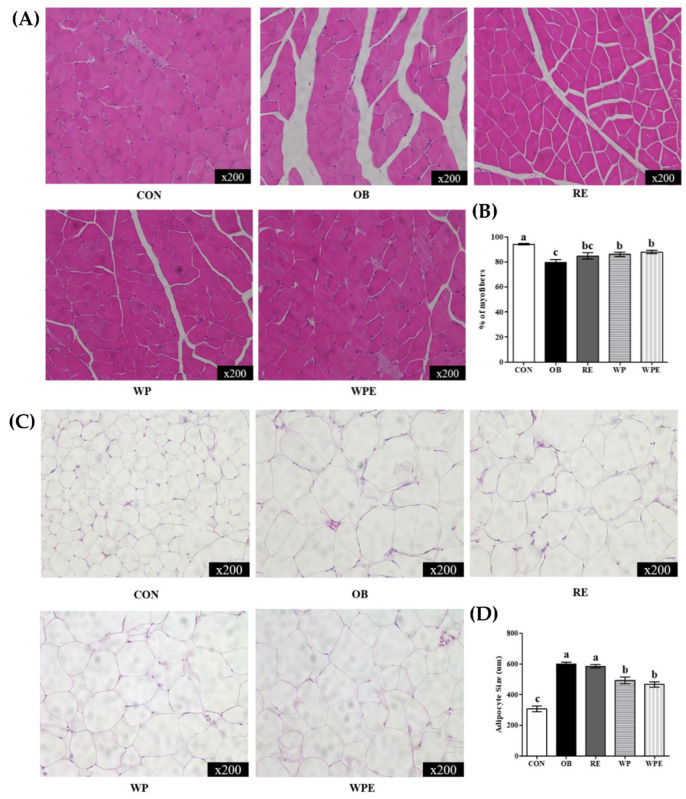
Effects of WP supplementation and RE on adipose tissue and skeletal muscle morphology (×200) in sarcopenic obese mice. (**A**) Adipose tissue morphology, (**B**) Mean adipocytes size (um), (**C**) Gastrocnemius morphology, and (**D**) Mean myofibers area (%). Values are means ± SEMs, *n* = 3–4. Mean values with the same superscript letter (a, b and c) are not significantly different (*p* < 0.05).

**Figure 4 nutrients-14-04402-f004:**
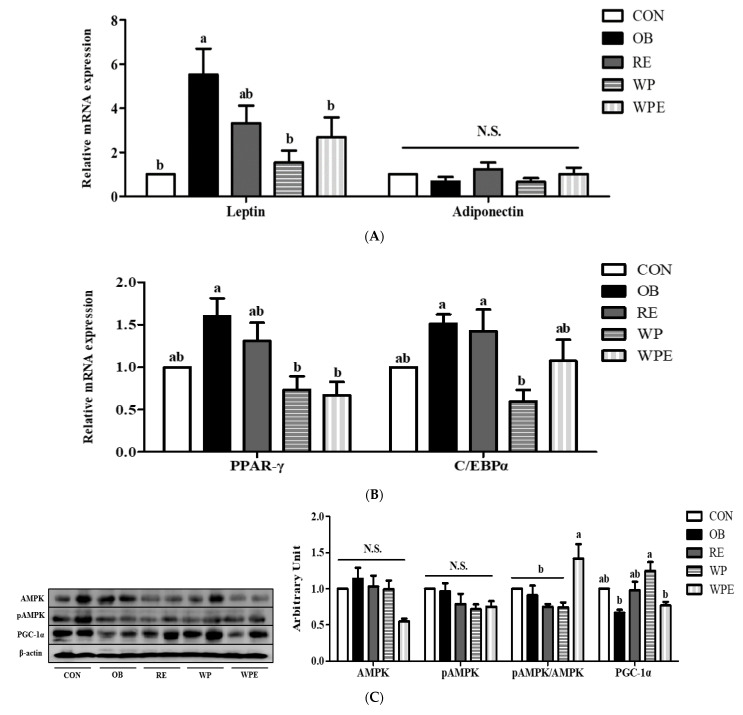
Effects of WP supplementation and RE on adipose tissue adipokines, lipid and energy metabolism in sarcopenic obese mice. (**A**) Adipokines (**B**) Lipid metabolism, and (**C**) Energy metabolism. Values are means ± SEMs, *n* = 6–8. Mean values with the same superscript letter (a and b) are not significantly different (*p* < 0.05). PPAR-γ, peroxisome proliferator-activated receptors γ; C/EBPα, CCAAT-enhancer-binding proteins α; AMPK, AMP-activated protein kinase; PGC-1α, proliferator-activated receptor-gamma coactivator 1-alpha. N.S.—Not significant.

**Figure 5 nutrients-14-04402-f005:**
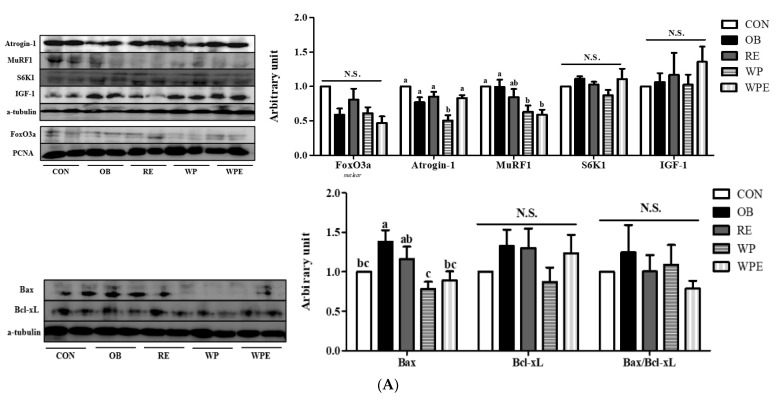
Effects of WP supplementation and RE on skeletal muscle protein and energy metabolism in sarcopenic obese mice. (**A**) Protein metabolism and (**B**) Energy metabolism. Values are means ± SEMs, *n* = 6–8. Mean values with the same superscript letter (a, b and c) are not significantly different (*p* < 0.05). Bax, bcl-2-like protein 4; Bcl-xL, B-cell lymphoma–extra large; FoxO3a, forkhead box O3; Atrogin-1, muscle atrophy F-box gene; MuRF1, E3 ubiquitin–protein ligase; S6K1, ribosomal protein S6 kinase beta-1: IGF-1, insulin-like growth factor I; Akt, protein kinase B. N.S.—Not significant.

**Table 1 nutrients-14-04402-t001:** Primer sequences used in the qRT-PCR assays.

Gene	Direction	Sequence (5′–3′)
GAPDH	Forward	AGG TTG TCT CCT GCG ACT
	Reverse	TGC TGT AGC CGT ATT CAT TGT CA
Leptin	Forward	GAG ACC CCT GTG TCG GTT C
	Reverse	CTG CGT GTG TGA AAT GTC ATT G
Adiponectin	Forward	TGA CGA CAC CAA AAG GGC TC
	Reverse	ACC TGC ACA AGT TCC CTT GG
C/EBPα	Forward	CGC AAG AGC CGA GAT AAA GC
	Reverse	CAC GGC TCA GCT GTT CCA
PPAR-γ	Forward	CGC TGA TGC ACT GCC TAT GA
	Reverse	AGA GGT CCA CAG AGC TGA TTC C

**Table 2 nutrients-14-04402-t002:** Effects of WP supplementation and RE on the diet intake, body, fat, and muscle weights, body composition, and muscle strength in sarcopenic obese mice.

Group		CON	OB	RE	WP.	WPE
**Body Composition (% of Total Mass)**						
Before 8 weeks of RE and WP (right after sarcopenic obesity induction)	Fat Mass	22.87 ± 0.90 ^b^	36.82 ± 1.36 ^a^	37.13 ± 1.49 ^a^	36.59 ± 1.56 ^a^	36.82 ± 1.28 ^a^
	Lean Mass	74.58 ± 0.87 ^a^	61.08 ± 1.34 ^b^	60.82 ± 1.47 ^a^	61.38 ± 1.49 ^a^	61.06 ± 1.22 ^a^
After 8 weeks of RE and WP	Fat Mass	25.22 ± 1.07 ^b^	40.51 ± 1.02 ^a^	39.89 ± 1.45 ^a^	37.61 ± 1.75 ^a^	36.70 ± 1.64 ^a^
	Lean Mass	72.11 ± 1.03 ^a^	57.41 ± 0.99 ^b^	57.91 ± 1.43 ^b^	60.27 ± 1.66 ^b^	61.02 ± 1.56 ^b^
Difference made during 8 weeks of RE and WP	Fat Mass	2.36 ± 0.83 ^abc^	3.69 ± 0.73 ^a^	2.76 ± 0.49 ^ab^	0.73 ± 0.66 ^bc^	0.01 ± 1.20 ^c^
	Lean Mass	−2.47 ± 0.81 ^bc^	−3.67 ± 0.70 ^c^	−2.90 ± 0.49 ^bc^	−0.81 ± 0.63 ^ab^	−0.20 ± 1.12 ^a^
**Body Weight (g)**						
Before 8 weeks of RE and WP (sarcopenic obesity induced)		31.56 ± 0.75 ^b^	40.46 ± 0.07 ^a^	40.25 ± 1.24 ^a^	41.16 ± 2.05 ^a^	40.01 ± 1.49 ^a^
After 8 weeks of RE and WP		31.88 ± 0.85 ^c^	45.09 ± 0.92 ^a^	42.16 ± 1.49 ^ab^	42.63 ± 2.66 ^ab^	39.62 ± 2.36 ^b^
Difference made during 8 weeks of RE and WP		0.33 ± 0.35 ^b^	4.63 ± 0.58 ^a^	1.90 ± 0.56 ^b^	1.47 ± 1.08 ^b^	−1.39 ± 1.02 ^b^
**Fat Weight (% B.W.)** (After 8 weeks of RE and WP)						
Total Fat		4.96 ± 1.43 ^c^	16.03 ± 2.51 ^a^	14.98 ± 3.04 ^ab^	13.06 ± 3.54 ^b^	12.53 ± 3.65 ^b^
Subcutaneous Fat		2.68 ± 0.96 ^b^	10.00 ± 2.53 ^a^	9.41 ± 2.16 ^a^	8.40 ± 3.33 ^a^	7.54 ± 2.83 ^a^
Epididymis Fat		1.93 ± 0.51 ^d^	5.72 ± 0.67 ^a^	5.21 ± 1.07 ^ab^	4.34 ± 0.68 ^c^	4.72 ±1.07 ^bc^
Brown Adipose Tissue		0.34 ± 0.09	0.32 ± 0.08	0.35 ± 0.06	0.32 ± 0.14	0.28 ± 0.06
**Muscle Weight (% B.W.)** (After 8 weeks of RE and WP)						
Total Muscle		2.20 ± 0.14 ^a^	1.60 ± 0.19 ^b^	1.67 ± 0.22 ^b^	1.70 ± 0.32 ^b^	1.80 ± 0.22 ^b^
Gastrocnemius		1.02 ± 0.07 ^a^	0.72 ± 0.07 ^c^	0.79 ± 0.09 ^bc^	0.77 ± 0.14 ^bc^	0.83 ± 0.08 ^b^
Quadricep		1.13 ± 0.09 ^a^	0.84 ± 0.12 ^b^	0.84 ± 0.14 ^b^	0.89 ± 0.18 ^b^	0.92 ± 0.16 ^b^
Soleus		0.05 ± 0.01 ^a^	0.04 ± 0.00 ^b^	0.04 ± 0.01 ^b^	0.04 ± 0.00 ^b^	0.05 ± 0.01 ^b^
**Muscle Strength (strength(N)/B.W(g))** (After 8 weeks of RE and WP)		0.034 ± 0.002 ^a^	0.019 ± 0.002 ^c^	0.025 ± 0.001 ^b^	0.018 ± 0.003 ^c^	0.027 ± 0.002 ^b^
**Daily Diet Intake (g)**		3.25 ± 0.11 ^a^	2.88 ± 0.08 ^b^	2.62 ± 0.08 ^bc^	2.70 ± 0.13 ^bc^	2.51 ± 0.08 ^c^

Values are means ± SEMs, *n* = 8~10. Mean values with the same superscript letter (a, b, c, and d) are not significantly different (*p* < 0.05).

## Data Availability

Not applicable.

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
