# Peer review of "Effects of Whey Peptide Supplementation on Sarcopenic Obesity in High-Fat Diet-Fed Mice"

_nutrients, 2022, doi:10.3390/nu14204402_

Round 1

Reviewer 1 Report

Comments 15th of September 2022

Manuscript: nutrients-1932391

Effects of Whey Peptide Supplementation and Resistant Exercise on Sarcopenic Obesity in High-Fat Diet-Fed Mice

Summary:

Although RCTs are done in humans combining protein and exercise in frail aged people are done, It is the effect of Whey peptide (a hydrolyzed form of whey protein with potential extra biological benefits) and exercise intervention on sarcopenic obesity is unknown. The authors use a sarcopenic obese mice model to study.

Major.

1.       It is unclear if the model used can be characterized as a sarcopenic obese model (no references) Missing phenotype data: e.g. muscle strength/physical activity, is 8 months old animals appropriate for the model?

2.       Design: The authors mentioned in the Introduction that WP has potential extra biological benefits. However, they have no control group with a “control” protein supplementation like Whey protein.

3.       Design: Unclear is the amount of Nitrogen in the diet compared to the amount of nitrogen in the supplementation.

4.       Statistics: Data expressed as mean[95%CI] should be more appropriate

5.       In M&M was mentioned food intake was monitored. However, no Diet Intake data are presented. Therefore it is unclear if WP effects on body composition were caused by less diet intake, which means less fat intake.

6.       Sarcopenic obesity is related to less muscle function/physical activity. No data are presented concerning resistance exercise performance. Therefore, it is unclear if the used model showed less muscle function/physical activity and the effect of WP on muscle function/physical activity is unclear.

Minor:

1.       Line11: ‘OB’ abbreviation means?

2.       Line 15- 20; Add some key data/statistics

3.       Figure 1. “3DW” is not explained.

4.       Statistics: Data expressed as mean[95%CI] should be more appropriate

5.       Real p-values are not given in Table 2. Therefore, please add them to the text. (line 149-159)

6.       Missing: Legend of Table 2 with explaining the symbols for statistics.

Author Response

Major.

  1. It is unclear if the model used can be characterized as a sarcopenic obese model (no references) Missing phenotype data: e.g. muscle strength/physical activity, is 8 months old animals appropriate for the model?

> Thank you for your comment. As you mentioned, sarcopenic obesity (SO) is defined by the increase in fat mass and the decrease in lean mass. Sorry for the missing muscle strength test (MST) performed in this study. We have included muscle strength data in Table 1. The CON group had significantly higher muscle strength compared to the high-fat diet-fed groups. Among high-fat diet-fed groups, only exercise groups showed stronger muscle strength. Although a lot of animals studies regarding SO have been conducted there are no clear criteria for the appropriate age of the SO animal model. According to previous studies, SO demonstrating an increase in fat mass and a decrease in lean mass was induced by a high-fat diet in different aged mice (the young, the middle aged, and the old). As we know, older adults have higher chance of sarcopenic obesity which is easily initiated in the middle age. Therefore, the animal model used in this study could be a good SO model for human application. A recently published study has demonstrated SO in the similar aged mice. <1>

<1> The ameliorating effects of metformin on disarrangement ongoing in gastrocnemius muscle of sarcopenic and obese sarcopenic mice. Biochim Biophys Acta Mol Basis Dis. 2022. 

2. Design: The authors mentioned in the Introduction that WP has potential extra biological benefits. However, they have no control group with a “control” protein supplementation like Whey protein.

> Thank you for your comment. A previous study demonstrated that whey peptide had superior antioxidant, anti-inflammatory, and cognitive function improvement effects to whey protein in aged mice <1>. It would have been better if there was a whey protein-supplemented group. However, we did not include the whey protein treatment group because there were already many interventions including exercise and whey peptide in this study. If we included the whey protein group, there would have been too many groups including whey protein with/without exercise. We will consider whey protein treatment as a positive control in the next experiments.

<1> The Antioxidant Effects of Whey Protein Peptide on Learning and Memory Improvement in Aging Mice Models. Nutrients. 2021. 

3. Design: Unclear is the amount of Nitrogen in the diet compared to the amount of nitrogen in the supplementation.

> Thank you for your comment. Despite the direct measurement of nitrogen amount, it was estimated. The high-fat diet we used contains 26.23g of protein / 100g of a high-fat diet. If nitrogen to protein conversion factor (6.38) is implemented, around 4.11g of nitrogen / 100g of a high-fat diet can be found. As the supplementation (whey peptide) is composed of amino acids, about 15.67g of nitrogen / 100g of whey peptide supplementation can be expected. Whey peptide-supplemented mouse (50g B.W.) is expected to have an additional intake of about 0.012 g of nitrogen per day.

4. Statistics: Data expressed as mean[95%CI] should be more appropriate

>  Thank you for your comment. Data were revised including mean [95%CI]. 

5. In M&M was mentioned food intake was monitored. However, no Diet Intake data are presented. Therefore it is unclear if WP effects on body composition were caused by less diet intake, which means less fat intake.

>Thank you for your comment. Sorry for the missing diet intake data. We have included diet intake data in Table 1.  

6. Sarcopenic obesity is related to less muscle function/physical activity. No data are presented concerning resistance exercise performance. Therefore, it is unclear if the used model showed less muscle function/physical activity and the effect of WP on muscle function/physical activity is unclear.

>Thank you for your comment. As you mentioned, sarcopenic obesity is closely related to lower muscle function. Sorry for the missing muscle strength test (MST) performed in this study. We have included muscle strength data in Table 1.

Minor:

Line11: ‘OB’ abbreviation means?

Line 15- 20; Add some key data/statistics

Figure 1. “3DW” is not explained.

Statistics: Data expressed as mean[95%CI] should be more appropriate

Real p-values are not given in Table 2. Therefore, please add them to the text. (line 149-159)

Missing: Legend of Table 2 with explaining the symbols for statistics.

Reviewer 2 Report

It is a study that tested the interventional effects of whey peptide intake and resistance exercise on various factors of body composition, morphology of tissues/organs, mRNA expression of adipokines and myokines. As described in the conclusion, it revealed that supplementation of whey peptide is effective anti-obesity. 

In addition, study design including Intervention protocols of supplements and exercise, biomarkers like adipokines and myokines, all things identified to reach the conclusion of this study are no error. 

However, this study confirmed the general fact that whey supplement has a positive effect on sarcopenic obesity. If the comparison group of whey peptide vs whey protein was added, it would have been possible to conclude that the benefit of whey peptide supplemental effect was a little more.

Minor correction requested..

The tables should describe all descriptors in each table.

Figure 3. (B) should be adipocytes and (D) should be myofibers.

Figure 5. Where is (C)?

Line 77: typo

Line 257: Figure 5B

Line 263: Figure 5C

Line 304: citation # should be 37

Author Response

  1. However, this study confirmed the general fact that whey supplement has a positive effect on sarcopenic obesity. If the comparison group of whey peptide vs whey protein was added, it would have been possible to conclude that the benefit of whey peptide supplemental effect was a little more.

=> Thank you for your comment. A previous study demonstrated that whey peptide had superior antioxidant, anti-inflammatory, and cognitive function improvement effects to whey protein in aged mice <1>. It would have been better if there was a whey protein-supplemented group. However, we did not include the whey protein treatment group because there were already many interventions including exercise and whey peptide in this study. If we included the whey protein group, there would have been too many groups including whey protein with/without exercise. We will consider whey protein treatment as a positive control in the next experiments

<1> The Antioxidant Effects of Whey Protein Peptide on Learning and Memory Improvement in Aging Mice Models. Nutrients. 2021

Reviewer 3 Report

In the manuscript “Effects of Whey Peptide Supplementation and Resistant Exercise on Sarcopenic Obesity in High-Fat Diet-Fed Mice” ,the authors performed a treatment of whey peptide and resistant exercise in 8-month-old C57BL/6J mice fed with high-fat diet. In this study, authors performed body weight, body composition and the protein and mRNA expression of key markers related to energy metabolism, protein deregardation, apoptosis and lipid metabolism in skeletal muscle and indicated that WP supplementation regardless of RE has potential anti-obesity and anti-sarcopenic effects in sarcopenic obesity. However, the manuscript requires a substantial revision for further publication.

1. Sarcopenia is associated with ageing and older individuals, and the sarcopenia phenotype is considered multifactorial, however, the mice that were used 12 month of age after HFD and WP or RE treatment. This roughly equates to middle age in humans to be demonstrating sarcopenic status with obesity. I would like to understand more why the age range was selected for the mice, and why older animals were not used to take into consideration the aging phenotype associated with the development of sarcopenia. This could also be explained in the manuscript. 

2. The preparation of the whey peptide should be described in more detail in Methods.

3. In Methods, how the researchers controlled for the consistency of resistance exercise effects of C57 mice in this study.

4. In the method, It seems that the ladder climbing with weight-bearing is a resistance exercise for mice with confounding factors, how researchers control the consistency of the effect of resistance exercise with home-made ladder in this study.

5. There are a lot of unclear descriptions in the figures, Tables and legend, and more notes should be added to the abbreviations below figure and Table. For example, what is the exact meaning of 3DW, X, and O in the table in Figure 1? There are many parts that are not described, almost in every figure and table. No clear enough description of the differences among several groups of in Figures and Tables.

6. The study suggested that consumption of whey protein hydrolysate exacerbates HF diet-induced body weight gain, impairs glucose homeostasis and increases inflammation in HFD male C57 mice (PMID: 33900305) .However, this result is contrary to your findings, What do you think is the reason for such different results?It should be elaborated in the discussion part.

7. There are many Figures in the text with inconsistent description and wrong order such as page 6 line 168 and 177 (Figure 3 A-B, Figure C-D). It must be serious to reconfirm the order of figures in the text.

8. In the results section, the expression of mRNA and protein for the control group should be normalized to 1.

9. In the results section, the number of mouse replicates should be given the exact n value, not n=3-4 (page 6 line 213) or n=6-8 (page 7 line 240), what is the reason for excluding mice in the groups ?

10.  There are some inconsistencies between the description of the results section and the figures, such as page 7 line230-233: There was no significant difference in the protein levels of peroxisome proliferator-activated receptor-gamma coactivator 1-alpha (PGC-1α) between the CON and OB groups. However, in Figure 4C showed the decreased protein expression of PGC-1αin OB group compared with the CON group. Is the decline of PGC-1α not significant in OB group compared with CON ? please check again. Figrue legend should be more clear and concise.

11.  The 8-12 month old mice are not aging mice, and it should be emphasized in discussion section that this model is an obese combined sarcopenia, not an aging-associated sarcopenia and obesity mouse model.

Round 2

Reviewer 1 Report

Summary:

Although RCTs are done in humans combining protein and exercise in frail aged people are done, It is the effect of Whey peptide (a hydrolyzed form of whey protein with potential extra biological benefits) and exercise intervention on sarcopenic obesity is unknown. The authors use a sarcopenic obese mice model to study.

Response to authors

1.       Author made remarks about the choice and phenotyping of the used mice model. Please add this information (including relevant references) in the introduction section.

2.       Author mentioned that data were revised including mean [95%CI]. However, data were not expressed as mean[95%CI] as suggested by the reviewer. Also in line 158 “expressed as means [95%CI] ± standard error of the mean (SEM)” is not correct.

3.       Design: Concerning nitrogen intake which is different between the groups/individual animals. I suggest making a separate table (or extend in Table 2) with all food intake details that potential can have an effect on your results (e.g. nitrogen, protein, carbohydrates, lipids). For instance, body composition is heavily dependent on the food intake and food composition.

4.       Design: Because a control protein supplementation is not added to this study, it is unclear if the effects of WP showed are related to more nitrogen intake or if there is a specific WP effect. This need to be discussed in the discussion section.

5.       Concerning the description results in the text (in result section and abstract), please add the real p-value.

E.g. line 15-17: “WP supplementation regardless of RE significantly suppressed the increasing fat mass (p=….) and decreasing lean mass (p=…….) and alleviated abnormal morphological changes in skeletal muscle and adipose tissue.

Reviewer 3 Report

The authors have addressed where possible the major queries that were submitted from the first review and included the relevant sections within the manuscript. The limitations are covered more accurately. Furthermore, authors have also added more detailed and accurate description in Figures and Tables and provided useful feedback to all comments.

Author Response

Thank you again for your useful critiques and constructive comments on our manuscript.